# Immunosuppressive Therapy of Antibody-Mediated aHUS and TTP

**DOI:** 10.3390/ijms241814389

**Published:** 2023-09-21

**Authors:** Kata Kelen, Orsolya Horváth, Éva Kis, Bálint Mikes, Péter Sallay, Zoltán Prohászka, Attila József Szabó, György S. Reusz

**Affiliations:** 1Bókay Street Unit, Department of Pediatrics, Semmelweis University, 1083 Budapest, Hungary; kelen.kata@med.semmelweis-univ.hu (K.K.); horvath.orsolya@med.semmelweis-univ.hu (O.H.); mikes.balint@med.semmelweis-univ.hu (B.M.); sallay.peter@med.semmelweis-univ.hu (P.S.); szabo.attila@med.semmelweis-univ.hu (A.J.S.); 2Department of Pediatric Cardiology, Gottsegen György Hungarian Institute of Cardiology, 1096 Budapest, Hungary; kiseva.mail@gmail.com; 3Research Laboratory, Department of Medicine and Hematology, Semmelweis University, 1083 Budapest, Hungary; prohaszka.zoltan@med.semmelweis-univ.hu; 4Pediatric Center, MTA Center of Excellence, Semmelweis University, 1083 Budapest, Hungary; 5ELKH-SE Pediatrics and Nephrology Research Group, 1052 Budapest, Hungary

**Keywords:** thrombotic microangiopathy, aHUS, TTP, immunosuppression, eculizumab

## Abstract

The recent classification of pediatric thrombotic microangiopathies (TMA) takes into consideration mechanisms of disease for guidance to targeted therapies. We present our experience with seven patients with antibody mediated atypical hemolytic uremic syndrome (aHUS) and thrombotic thrombocytopenic purpura (TTP). Five children had aHUS with antibodies against complement factor H (CFH-ab) and two with TTP with antibodies against metalloproteinase ADAMTS13. In the aHUS cases diagnosed and treated before the eculizumab era, CFH-ab was detected using the ELISA assay. Mutational analysis of selected complement genes was performed. TTP was diagnosed if, in addition to microangiopathic hemolytic anemia and thrombocytopenia, ischemic organ involvement and severe deficiency in ADAMTS13 activity were present. Treatment protocol consisted of plasma exchanges (PE) and steroid pulses, followed by the combination of cyclophosphamide and rituximab to achieve long-term immunosuppression. Four patients with CFH-ab and the TTP patients with ADAMTS13 antibodies came into sustained remission. After a median follow-up of 11.7 (range 7.7–12.9) years without maintenance therapy, no disease recurrence was observed; nevertheless, six patients, two had hypertension and two had proteinuria as a late consequence. One patient, with late diagnosis of CFH-ab and additional genetic risk factors who was treated only with PE and plasma substitution, reached end-stage renal disease and was later successfully transplanted using eculizumab prophylaxis. In the cases of antibody-mediated TMAs, PE and early immunosuppressive treatment may result in sustained remission with preserved kidney function. Further data are needed to establish optimal treatment of anti-FH antibody-associated HUS.

## 1. Introduction

Hemolytic uremic syndrome (HUS) belongs to the group of thrombotic microangiopathies (TMAs). It is defined by hemolytic anemia, thrombocytopenia and kidney failure. The typical diarrhea-associated form of HUS is caused by the exotoxins of certain bacteria, most frequently enterohemorrhagic *Escherichia coli*. By contrast, the atypical form of HUS (aHUS) is a consequence of deficiency in the regulatory proteins of the alternative complement pathway [1,2]. Sixty to seventy percent of aHUS patients carry autoantibodies against factor H (FH) or mutations in the genes encoding for complement factor H proteins (CFH), complement factor H-related protein 5 (CFHR5), factor I (CFI), membrane cofactor protein (CD46), C3 (C3), factor B (CFB) or thrombomodulin (THBD). In such cases, uncontrolled complement activation and vascular injury of aHUS may be triggered by a known or not yet recognized factor, although some cases have persistent complement activation and vascular injury that is punctuated by periods of exacerbation [3,4,5].

TMA may also be caused by the decreased activity of ADAMTS13 (A disintegrin and metalloproteinase with a thrombospondin type 1 motif, member 13), responsible for the cleavage of the von Willebrand factor. In this form, TMA results from excessive systemic platelet aggregation caused by the accumulation of ultra-large multimers of von Willebrand factor in the plasma. Severe ADAMTS13 deficiency results either from a genetic defect or from polyclonal autoantibodies in acquired thrombotic thrombocytopenic purpura (TTP) [6,7].

The detailed presentation of the TMA classification is beyond the scope of this paper; for details, please refer to current guidelines and review articles for a comprehensive classification of TMAs [1,8].

The understanding of pathways involved in the mechanism of aHUS has allowed for the development of novel treatment options targeting terminal complement activation using an anti-C5 antibody (eculizumab). Subsequent studies based on eculizumab treatment have profoundly changed our view on the treatment algorithm already proposed in the 2009 consensus paper on the management of aHUS [2,5]. Eculizumab is now considered as first-line treatment, if available, in all pediatric aHUS cases [2,9]. If eculizumab is not (or not immediately) available, plasma exchange (PE) (or plasma infusion (PI) if PE is not accessible) should be started as recommended [10].

In cases of antibody-mediated aHUS and TTP, removing antibodies using PE and inhibiting their production by immunosuppression has proven an effective treatment option [11,12,13]. The importance of the topic is reflected by the fact that a separate treatment algorithm is proposed for anti-FH antibody-associated HUS in the 2015 consensus paper [5].

Herein, we present a small series of aHUS and TTP cases caused by antibodies to factor H in five cases (diagnosed between 2008–2013 before the availability of eculizumab in Hungary) and two caused by antibodies against ADAMTS13 (diagnosed in 2010 and 2015). All patients were diagnosed in one center, treated with plasma infusion or plasma exchange and achieved a first remission. All but one were also treated initially using immunosuppression (IS) comprising steroids, cyclophosphamide and finally rituximab (RTX). All patients treated with early IS achieved sustained remission, without maintenance therapy.

## 2. Results

The clinical characteristics of our patients at diagnosis are presented in Table 1. As our diagnostic workup was incomplete at the time of the first patient (patient 1), the presence of FH antibodies was not tested, and the patient was categorized as factor H deficient. PE was used to achieve remission, which was subsequently continued with plasma infusions. The patient had a remitting–relapsing course, requiring three repeated separate PE courses, with slowly worsening kidney function ultimately reaching end stage chronic kidney disease (CKD5D) after 204 days where continuous ambulatory peritoneal dialysis (CAPD) was initiated. She had one late recurrence of HUS after 2 years on CAPD, when FH antibodies were first investigated and detected. PE was reintroduced, and three methylprednisolone pulses were given, followed by a one-month steroid tapering. Finally, due to uncontrolled hypertension, bilateral nephrectomy was performed 58 months after disease onset. Complete genetic workup (including the CFH, CFHR5, CFI, CD46, C3, CFB and THBD genes) identified two missense mutations in heterozygous form in the THBD (c.1456G > T, p.D486Y) and CFHR5 (c.329T > C, p.V110A) genes in this patient, both leading to amino acid changes. Previously, the identified D486Y variant THBD protein was shown to exhibit defects in suppressing activation of the alternative complement pathway through factor I-mediated C3b inactivation in vitro, suggesting its role in the pathogenesis of aHUS [14].

Up to now, no studies have been performed on the possible functional effect of the V110A variant CFHR5 protein, while the predictions of the applied in silico tools are ambiguous (two “neutral” and two “possibly damaging” predictions). Furthermore, patient 1 carried the CFH c.-331C > T polymorphism in heterozygous form (reported as a risk factor of aHUS) and was also found to be homozygous for a common deletion of the CFHR1 and CFHR3 genes, which is associated with the presence of anti-factor H autoantibodies.

In patients 2 to 5, the diagnosis of aHUS was made within 4 days after the clinical diagnosis of HUS. The presence of FH antibodies was verified within 1 week after the clinical diagnosis. Patients 3–5 were found to be homozygous for the deletion of the CFHR1 and CFHR3 genes, whereas patient 2 carried the homozygous deletion of CFHR1 along with the heterozygous deletion of CFHR3.

The diagnosis of TMA in patient 6 was made 6 weeks after initial presentation based on neurological involvement accompanied by anemia and sustained thrombocytopenia and only slightly reduced kidney function. In contrast, in patient 7, signs of hemolytic anemia and thrombotic microangiopathy were associated with more severe kidney involvement. In both cases, profoundly suppressed ADAMTS13 activity and the presence of ADAMTS13 antibodies were the clue to the diagnosis.

In these two cases of antibody-mediated TTP, the same therapeutic protocol was used as in patients 2 to 5 with FH antibodies. Daily PE treatment was initiated and continued every other day until clinical remission was achieved, after which cyclophosphamide and finally rituximab were used to maintain sustained remission.

Relevant laboratory data at the end of the IST are shown in Table 2.

All patients achieved hematological and clinical remission at the end of the immunosuppressive treatment (IST). All patients had normal eGFR, kidney function and no proteinuria.

Complement activity and factor H level returned to normal in patients 1 to 5 following PE treatment. In patients 3 to 5 receiving IST, the remission was sustained. FH antibody levels also decreased and remained low throughout the observational period. In patient 2, treated with monthly PI for 6 months, the FH antibody level decreased more slowly, reaching below 200 AU/mL on day 525.

ADAMTS13 activity normalized, and antibodies disappeared during PE in patient 6 with sustained remission for about 3 years after completely ceasing all IST. However, antibodies reappeared and ADAMTS13 activity decreased 44 months after discontinuing RTX therapy. Although no clinical recurrence occurred, it was decided to repeat RTX. Following two infusions, ADAMTS13 activity increased and was normalized after 2 months.

The proportion of CD20-bearing cells fell below 1% in all patients following RTX treatment. The reappearance of CD20-positive cells did not translate into an increase in FH antibodies. Indeed, at time of writing, patients 2 to 5 are in remission despite the normal CD20 cell count.

In patient 6, the number of CD20-positive cells increased well before the reappearance of ADAMTS13 antibodies. Thus, the monitoring of CD20-positive cells does not appear to be a sensitive method for predicting the reoccurrence of antibody-mediated TMA [16,17,18].

Data from the end of follow-up are shown in Table 3.

At time of writing, patient 4 and 7 have a 0.5–1 g/day proteinuria; patient 4 is treated with a combination of perindopril, indapamide and amlodipine; and patient 5 is treated with ramipril for hypertension. After a median follow-up of 11.8 (range 7.7–15.2) years, all but one patient (patient 1) had functioning native kidneys. Three out of six patients were in stage 2 chronic kidney disease (CKD); the others had normal kidney function. Two of six patients were hypertensive, and two of six patients had proteinuria. Patient 1 was transplanted using eculizumab prophylaxis after 7 years of renal replacement therapy (initially peritoneal dialysis, switched to hemodialysis due to technical difficulties); at the last visit, 6 years after kidney transplantation, she was in complete clinical and hematological remission. She is on ongoing C5 blockade and was switched to long-acting ravulizumab when it became available. Her eGFR is normal (>90 mL/min/1.73 m^2^); she is normotensive with no proteinuria.

At the last visit, the complement profile was reassessed in patients 1–5; there was no evidence of complement consumption or complement activation.

## 3. Discussion

This small series of patients with antibody-induced aHUS and TTP reveals the role of early antibody testing in the acute phase of the disease, since a positive result enables additional treatment options [6,7,11,12,18]. PE and early IST may result in sustained remission with preserved kidney function in aHUS due to FH antibodies as well as in TTP resulting from antibodies against ADAMTS13. It also points to the necessity of a complete laboratory workup in the case of unexpected or unexplained recurrences, such as in the case of patient 1, where in addition to the presence of FH antibodies, additional potential predisposing factors occurred (see below).

The mechanism leading to antibody formation is different in antibody-mediated HUS and TTP. Our anti-CFH cases had genomic deletion of CFHR3/CFHR1. In the case of our patients with acquired TTP, we could not detect the appearance of any other autoantibody during the follow-up.

Although C5 inhibitor therapy is the first line treatment in childhood aHUS if available [12], in FH antibody-mediated aHUS, immunosuppressive treatment may ensure the elimination of antibodies and may allow for C5 inhibitor treatment to be suspended. In antibody-mediated TTP, plasma exchange and IST are currently still the standard therapeutic options in pediatric population [19].

Of our aHUS patients diagnosed before the availability of eculizumab, four patients can be classified as patients in complete and sustained remission. In the long term, however, signs of CKD developed such as decrease of kidney function (CKD2 in 3/4 of the cases), hypertension (2/4) and proteinuria (1/4). In fact, this is presumably a result of the initial kidney damage as the underlying aHUS disease is in remission. One patient with delayed diagnosis and a more complex genetic background achieved CKD5D and was successfully transplanted years later under C5 inhibition protection.

As for the two TTP cases, regular ADAMTS13 activity monitoring showed normal values; their blood pressure and eGFR were normal; one patient developed proteinuria.

All of these data highlight the importance of long-term follow-up, even in the case of initially successful treatment. For this reason, our patients are regularly checked (including possible recurrence of antibodies, kidney function check-up including GFR and proteinuria tests). However, even 11.7 years (range 7.7–12.9) after disease onset, they show no signs of renal or extrarenal reactivation of their disease.

Due to the small number of patients, it is difficult to compare the clinical features of our patients with the series of children with FH antibodies published. [11,12] PE and IST resulted in sustained remission, normal FH activity and disappearance of the high titers of FH antibodies in patients 2 to 5, without the need of maintenance immunosuppression in contrast to the large Indian series [5,11].

The cause of the first patient’s disease was complex: multiple abnormalities affecting the complement system could be detected. She was found to be homozygous for a common deletion of the CFHR1 and CFHR3 genes, associated with the presence of anti-factor H autoantibodies. Further, we identified a previously described [3,20,21,22] high-risk mutation in the THBD gene (c.1456G > T, p.D486Y) alleged to contribute to the pathomechanism of aHUS. Previously described mutations that negatively affect the function of thrombomodulin occur in approximately 5% of patients with aHUS. [20] Patient 1 also carried a substitution in CFHR5 (c.329T > C, p.V110A) although the role of this rare variation in the pathomechanism of aHUS is still unknown. This case history underlines the necessity of detailed workup of all aHUS patients, since multiple risk factors and variations may occur in a given patient, which may also influence peritransplantation management of the planned kidney transplant of the child.

Patient 6 and 7 had TTP due to antibodies to ADAMTS13 causing severe deficiency (<5%) in ADAMTS13 activity. ADAMTS13 deficiency-associated TTP needs to be ruled out in patients suspected of having aHUS [6].

In the present series, steroid pulses were used as first-line immunosuppression combined with PE. Cyclophosphamide was used to achieve long-term effects, while RTX was used as specific anti-B cell treatment in a nonspecific attempt to block the reappearance of the FH antibodies.

Our data on the use of cyclophosphamide is in accordance with those of Sana G et al., who presented a case series of antibody-mediated aHUS cases treated similarly with cyclophosphamide, resulting in rapid and sustained remission up to 6 years, 4 years and 4 months without any maintenance therapy [12].

The rationale for combining RTX with cyclophosphamide is based on the differential effect of these indications across the B-cell lineage. RTX is a B-cell-depleting monoclonal antibody that binds to cluster of differentiation (CD) 20 expressed on the surface of B-cells while it has little direct effect on the antibody-producing plasmablasts and plasma cells that do not express CD20 [23,24].

The addition of cyclophosphamide can target autoantibody-producing plasmablasts and short-lived plasma cells, targeting cells not affected by anti-CD20 antibodies [25]. In our study with regard to the combined treatment the number of cyclophosphamide shots (and its cumulative dose) did not reach the dose used in immune-mediated glomerular diseases (systemic lupus erythematosus vasculitis, rapidly progressive glomerulonephritis) [26].

## 4. Materials and Methods

Seven patients were diagnosed with antibody-associated TMA between 2008 and 2013. Five had FH antibody-mediated aHUS while two patients were diagnosed as having anti-ADAMTS13 associated TTP.

The diagnosis of aHUS was based on the presence of acute Coombs-negative microangiopathic hemolytic anemia with schistocytes, thrombocytopenia (<150 × 10^9^/L) and acute kidney injury (serum creatinine level > normal values for age and/or estimated glomerular filtration rate (eGFR) < 90 mL/min/1.73 m^2^ and/or proteinuria > 0.5 g/day) at any time during disease course. GFR was estimated by the modified Schwartz formula [15]. Stool culture and detection of Shiga toxins in stool were negative.

Thrombotic thrombocytopenic purpura (TTP) was diagnosed if, in addition to microangiopathic hemolytic anemia and thrombocytopenia (defined as above), ischemic organ involvement (renal, central nervous system (CNS) and severe deficiency (<10%) in ADAMTS13 activity were present [27].

### 4.1. Determination of Complement Parameters and ADAMTS13 Activity

Functional assessment of the alternative pathway was performed with a commercial kit (Wieslab AP enzyme-linked immunosorbent assay (ELISA) KIT, Malmö, Sweden) [28], total classical pathway activity by sheep-erythrocyte hemolytic test, C3 was measured by immunoturbidimetry, factor H antigen by sandwich-ELISA, and factors C4, B and I with radial immune diffusion. ADAMTS13 activity levels were determined using the fluorigenic substrate FRET-VWF73 (Peptides International, Louisville, KY, USA). The presence of anti-ADAMTS13 inhibitors (inhibitory antibodies) was determined by a functional assay in which the patient’s sample was mixed with a pooled normal human sample in a 1:1 ratio and incubated for 2 h at 37 °C before measuring the activity as described above. In the absence of inhibitors, the mixed sample’s activity is above 50%, whereas an activity below 35% indicates the presence of anti-ADAMTS13 inhibitors. Details of the above laboratory methods and normal values have been described elsewhere [16,17].

IgG autoantibodies against factor H were determined via homemade ELISAs. In detail, microtiter plates were coated with 1 microgram/mL purified human Factor H (Calbiochem, Darmstadt, Germany) in bicarbonate buffer overnight. After washing, wells were incubated with diluted test sera and bound IgG measured with horseradish-peroxidase conjugated anti-human IgG antibodies (DAKO, Glostrup, Denmark). Titers of positive samples were expressed as arbitrary units per mL (AU/mL) and calculated using a calibration curve based on serial dilutions of a reference positive plasma (used as 1000 AU/mL), obtained from Marie-Agnés Dragon-Durey [IROD-UJ]. The positive threshold (110 AU/mL) was calculated by the mean + 2 SD of 200 individual healthy donor plasma values (N = 100) based on in-house measurements. Exclusion of false positive results was done as part of laboratory quality control.

Mutational analysis of selected complement genes and regulators (CFH, CFHR5, CFI, CD46, C3, CFB, THBD), as well as the determination of copy number variations in the genes encoding the complement-factor H related proteins (CFHR1, CFHR2, CFHR3, CFHR5) were carried out as described previously [18].

### 4.2. Treatment Protocol

The treatment protocol consisted of daily PEs (with a calculated 1.5 times plasma volume) continued every other day until reaching hematological and clinical remission, followed by weekly PIs.

Immunosuppressive treatment was administered in Patients 2–7 in the form of methylprednisolone, cyclophosphamide and rituximab.

Methylprednisolone was given during PE therapy following the PE sessions (3 × 500 mg/1.73 m^2^ every other day in 100 mL of saline), after which daily oral methylprednisolone (0.8 mg/kg) was prescribed, with weekly tapering and discontinuation 40–60 days after clinical presentation.

After at least ≥25% decrease in serum creatinine remission and terminating plasma exchange, Patients 2–7 were treated with cyclophosphamide pulses (750 mg/m^2^ i.v. three times every two weeks, in each case in 500 mL saline solution), followed by hyperhydratation and mesna infusion to decrease the risk of bleeding from the bladder. Finally, rituximab (375 mg/m^2^ weekly) 4 times was given 1 month after the last cyclophosphamide induction. PI was stopped at the end of the IST.

Supportive treatment with weekly, then bi-weekly PI was continued in Patient 1 (who was not treated initially with IST—see below for further details).

Patients were classified as achieving complete thrombotic microangiopathy response if they reached haematologic normalization and a ≥25% decrease in serum creatinine persisting for ≥4 weeks. A haematologic normalization was defined as platelet count normalization (≥150 × 10^9^/L) and LDH normalization (˂1.5 times the age-matched upper limit of normal). Each normalization should be maintained for ≥4 weeks [29].

### 4.3. Ethics

Retrospective data collection for this study was performed in accordance with the Declarations of Helsinki and approved by the institutional Ethics Committee on Human Research.

## 5. Conclusions

Due to the small number of cases, we cannot draw far-reaching conclusions either in terms of renal outcome or remission; it does not allow for detailed statistical analysis and does not allow for insight into the detailed mechanism of immunosuppressive treatment. Several factors (e.g., dehydration) may have influenced the initial kidney involvement, but this cannot be ascertained in retrospect. For example, the initial low GFR in patient 7 with TTP is unusual and may also have an extrarenal origin (dehydration before admittance to the intensive care unit), which is supported by the rapid improvement in kidney function.

The strength of the study lies in the long-term follow-up, demonstrating long-term remission that did not require further immunosuppression. No complications of immunosuppression (infections, eventual malignancies) were detected.

In the era of the availability of C5 blockade, the treatment we used can be a supplement to C5 blockade for patients with antibodies against complement factor H, which may also allow for the subsequent suspension of C5 inhibition.

## Figures and Tables

**Table 1 ijms-24-14389-t001:** Initial clinical and laboratory features at the diagnosis of antibody-mediated aHUS and TTP.

Initial Clinical and Laboratory Features																	
Patient Number	Dg	M/F	Age	Initial Clinical Features	WEIGHT (kg)	Height (cm)	Hgb (g/L)	Plt (G/L)	Cr micro- mol/L	eGFR (mL/min/1.73 m^2^)	LDH (U/L)	AP (%)	C3 (g/L)	FH (mg/L)	FH Ab (AU/mL)	ADAMTS 13 (%)	ADAMTS 13 IA
Normal Range							130–165 for M, 115–150 for F	150–400	40–90	>90	˂1.5 Times the Age-Matched UPN		0.9–1.8	250–880	<110	67–151	POS: <35% NEG: >50%
1	aHUS	F	7.4	petechiae, vomitus	19	122	57	33	207	29	2233	2	0.67	<30	NA	93	NM
2	aHUS	F	8.0	edema, petechiae, vomitus, weakness	32	139	105 *	23	105	64	1894	64	0.62	98	2221	44	NM
3	aHUS	M	6.4	headache, vomitus, petechiae	26	130	91	21	86	74	2946	3	0.54	70	10,067	80	NM
4	aHUS	M	8.4	vomitus, weakness	29	130	106 *	44	159	40	4300	49	0.69	119	2190	76	NM
5	aHUS	M	10.9	dizziness, diarrhea (2 weeks before HUS)	34	144	80	41	503	18	7078	82	0.79	60	308	39	NM
6	TTP	F	12.8	headache, dizziness, blurred vision, nausea	31	151	102 *	37	91	80	868	81	1.32	203	<110	0	POS
7	TTP	F	16.0	fever, dark urine, petechiae	45	167	126	8	272	22	4031	111	0.98	415	<110	2	POS

ADAMTS 13: A disintegrin and metalloproteinase with a thrombospondin type 1 motif, member 13; ADAMTS 13 IA: inhibitor activity against ADAMTS 13; aHUS: atypical hemolytic uremic syndrome; AP: complement alternative pathway; C3: complement factor 3; Cr: creatinine; Dg: diagnosis; eGFR: estimated glomerular filtration rate (mL/min/1.73 m^2^) according to Schwartz [15]; FH: complement factor H; FH Ab: antibodies against complement factor H; Hgb: hemoglobin; HUS: hemolytic uremic syndrome; LDH: lactate dehydrogenase; M/F: male/female; NA: not available; NM: not measured; Plt: platelets; POS: positive; TTP: thrombotic thrombocytopenic purpura; UPN: upper limit of normal. * Transfused before admission.

**Table 2 ijms-24-14389-t002:** Laboratory data at the end of immunosuppressive treatment.

Laboratory Data at the End of Immunosuppressive Treatment														
Patient number	Dg	Day	Hgb (g/L)	Plt (G/L)	Cr (micromol/L)	eGFR (mL/min/1.73 m^2^)	LDH (U/L)	AP (%)	C3 (g/L)	FH (mg/L)	FH Ab (AU/mL)	ADAMTS 13 (%)	ADAMTS 13 IA	CD20 (%)
Normal Range			130–165 for M, 115–150 for F	150–200	40–90	>90	˂1.5 Times the Age-Matched UPN	70–125	0.9–1.8	250–880	<110	67–151	POS: <35% NEG: >50%	
1 *	aHUS	NA	NA	NA	NA	NA	NA	NA	NA	NA	NA	NA	NA	NA
2	aHUS	96	130	305	65	>90	698	94	1.44	186	665	NM	NM	<1
3	aHUS	89	125	308	68	>90	449	78	1.16	160	177	NM	NM	<1
4	aHUS	121	126	400	61	>90	504	77	0.87	190	88	NM	NM	<1
5	aHUS	192	135	340	48	>90	470	87	1.06	219	147	NM	NM	<1
6	TTP	154	129	374	73	>90	394	NM	NM	NM	NM	77	NEG	<1
7	TTP	143	148	375	53	>90	315	99	1.25	523	NM	63	NEG	<1

ADAMTS 13: A disintegrin and metalloproteinase with a thrombospondin type 1 motif, member 13; ADAMTS 13 IA: inhibitor activity against ADAMTS 13; aHUS: atypical hemolytic uremic syndrome; AP: complement alternative pathway; C3: complement factor 3; CD20: B-lymphocyte antigen CD20; Cr: creatinine; Dg: diagnosis; eGFR: estimated glomerular filtration rate (mL/min/1.73 m^2^) according to Schwartz [15]; FH: complement factor H; FH Ab: antibodies against complement factor H; Hgb: hemoglobin; LDH: lactate dehydrogenase; M/F: male/female; NA: not available; NM: not measured; Plt: platelets; POS: positive; TTP: thrombotic thrombocytopenic purpura; UPN: upper limit of normal. * Patient 1 did not receive immunosuppressive treatment before the development of end stage kidney failure.

**Table 3 ijms-24-14389-t003:** Clinical and laboratory characteristics of patients with antibody-mediated aHUS and TTP after long-term follow-up.

Clinical and Laboratory Features after Long-Term Follow-Up															
Patient Number	Dg	Age at Dg	Age at Last Follow-Up	Follow-Up (Year)	Clinical Features	Current Complement Blocking Therapy	Proteinuria	Hyper-Tension	Cr micro- mol/L	Stage of Kidney Disease	LDH (U/L)	C3 g/L	FH mg/L	FH Ab (AU/mL)	ADAMTS 13 (%)
Normal Range									40–90	eGFR < 90	˂1.5 Times the Age-Matched UPN	0.9–1.8	250–880	<110	67–151
1	aHUS	7	23	15.2	KTX 2017	Yes	No	No	54	eGFR > 90 after KTX	239	0.9	424	73	NM
2	aHUS	8	20	12.3	Treatment for astma bronchiale	No	No	No	79	CKD2	157	1.38	322	82	85
3	aHUS	6	18	11.8		No	No	No	93	CKD2	168	0.79	395	210	NM
4	aHUS	9	20	11.6		No	Yes	Yes	96	CKD2	165	1.19	583	50	NM
5	aHUS	11	21	10.1		No	No	Yes	60	eGFR > 90	209	1.19	415	102	106
6 *	TTP	13	26	12.9		No	No	No	61	eGFR > 90	281	NM	NM	NM	100
7	TTP	16	24	7.7		No	Yes	No	65	eGFR > 90	217	NM	NM	NM	74

Ab: antibody; aHUS: atypical hemolytic uremic syndrome; CKD: chronic kidney disease; Cr: creatinine; Dg: diagnosis; FH: factor H; eGFR: estimated glomerular filtration rate (mL/min/1.73 m^2^) according to Schwartz [15]; KTX: kidney transplantation; LDH: lactate dehydrogenase; NM: not measured; TTP: thrombotic thrombocytopenic purpura; UPN: upper limit of normal. * Case 6 received repeated rituximab therapy at 44 months for reappearance of ADAMTS13 antibodies and decrease of ADAMTS13 activity. After the repeated rituximab therapy ADAMTS13 activity elevated back immediately (168%).

## Data Availability

Data is contained within the article.

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
