# Peer review of "Immunosuppressive Therapy of Antibody-Mediated aHUS and TTP"

_ijms, 2023, doi:10.3390/ijms241814389_

Round 1

Reviewer 1 Report

In this manuscript, entitled “Immunosuppressive therapy of antibody-mediated thrombotic microangiopathies”, the authors describe long-term remission following pulse steroids, cyclophosphamide and rituximab in five children of atypical HUS with positive complement factor H (CFH) antibodies and two children of TTP with ADAMTS13 antibodies. There are several serious issues in the manuscript that invalidate or do not support the optimistic conclusion long term remission without maintenance therapy. Together these flaws preclude a complete review of the manuscript.

Most critically, the definition of remission for aHUS and TTP reflects a lack of the knowledge of the full spectra of the two disorders. Atypical HUS may present with progressive renal injury, with or without dysfunction of other organs such as the brain, eye, chest/lung, heart and abdomen/gastrointestinal tract in the absence of overt microangiopathic hemolysis and thrombocytopenia. In this regard, it is strange that the authors are not alarmed that all four eligible aHUS patients (excluding case 1) had deterioration of their kidney function, new proteinuria, and/or new hypertension following their strategy of intervention. With presently available effective therapy, such outcomes are neither optimal nor acceptable.   

Overt relapse of acquired TTP is often heralded by the progressive decline of ADAMTS13 activity. Case 6 showed this course. Acquired TTP with ADAMTS13 activity persistently below 10% may also be complicated with progressive organ injury, most commonly of their brains and kidney without overt evidence of microangiopathic hemolysis and thrombocytopenia. Hence, follow-up of acquired TTP should include both monthly ADAMTS13 activity and assessment of organ functions.     

Other issues of serious concern include inappropriate title, conflicting definition of diagnostic terms and disease classification, inadequate characterization of the cases, inappropriate inclusion of cases and inadequate description of the follow-up data.

The tile is inappropriately broad; the case series does not encompasses all disorders that may fall in the category of “antibody-mediated thrombotic microangiopathies”. Furthermore, it inputs the concept that aHUS with anti-FH and TTP with anti-ADAMTS13 are equivalent in their autoimmunity, ignoring that possibility that they may be quite different in their pathogenesis of autoimmunity. Most anti-CFH cases have genomic deletion of CFHR3/CFHR1 whereas many cases of acquired TTP have other autoimmune antibodies suggesting a genetic predisposition to autoimmunity.

Diagnostic terms:

“Thrombotic microangiopathy”: In the literature, thrombotic microangiopathy is often used indistinguishably for the clinical syndrome of thrombocytopenia and the pathological lesion of thrombotic microangiopathy. However, these are two different diagnostic terms that are not interchangeable. The authors should clearly indicate their definition of “thrombotic microangiopathy”.

“By contrast, the atypical form of HUS (aHUS) is predominantly due to complement dysregulation triggered by a known or not yet recognized factor.” (Page 4)

This sentence is conceptually flawed and incorrect. In aHUS, complement dysregulation is a consequence of deficiency in the regulatory proteins of the alternative complement pathway, due to genetic mutations or antibodies. In such cases, uncontrolled complement activation and vascular injury of aHUS may be “triggered by a known or not yet recognized factor”, although some cases have persistent complement activation and vascular injury that is punctuated by periods of exacerbation.

“The diagnosis of aHUS was based on the presence of acute Coombs-negative microangiopathic hemolytic anemia with schistocytes, thrombocytopenia (<150 × 10_9/l) and acute kidney injury (serum creatinine level > normal values for age and/or estimated glomerular filtration rate (eGFR) < 90 ml/min/1.73 m2 and/or proteinuria > 0.5 g/day) at any time during disease course. GFR was estimated by the modified Schwartz formula. [14] Serum antibodies against lipopolysaccharides, stool culture and detection of Shiga toxins in stool were negative.” (Page 6)

This definition includes disorders that are unrelated to dysregulation of the alternative complement pathway. Positive complement factor H antibody test should be included in the case definition for this manuscript, but only after its false positivity is excluded. On the other hand, the authors should explain why antibodies of “lipopolysaccharides” are relevant in the case definition.

The diagnostic criteria for TTP encompass both acquired and hereditary TTP. For this manuscript, positive ADAMTS13 inhibitor test should be included in the case definition. ADAMTS13 antibody test may be a substitute, but only after its false positivity is excluded.  

Inappropriate inclusion of cases:

Case 1 should not be included in the case series. Although she had positive factor H antibody, there is no evidence that she benefited from the immunosuppressive therapy. Uncontrolled complement activation and renal injury of aHUS cease to occur when the kidney function is completely lost. After kidney transplantation, she required complement blocking therapy to prevent aHUS relapse, suggesting that she likely had persistently defective regulation of the alternative complement pathway.

Case 6 received repeated rituximab therapy at 44 months for reappearance of ADAMTS13 antibodies and decrease of ADAMTS13 activity. This caveat should be noted in Table 3. Furthermore, the duration of normalized ADAMTS13 activity is within the range expected following rituximab mono-therapy.

Case 7: The responses of ADAMTS13 activity and antibody should be provided to confirm her response to the treatment.  

Inadequate characterization of the cases:

Cases 1 and 3 have very low AP%, which is not an expected feature of aHUS. The causes of low AP% should be determined.

Case 7: Acquired TTP is not expected to decrease the eGFR to 22. Either the diagnosis of acquired TTP was incorrect or the patient had another cause of renal failure.        

Laboratory tests:

Tables 2 and 3: Reference ranges should be provided for all the tests listed.

For the homemade “IgG autoantibodies against factor H” the number of normal cases tested to establish the normal range should be indicated.

The authors should explain why FH Ab is reported as <110 for patient 6, but as 8 for patient 7.

ELISA tests may yield false positive results for ADAMTS13 Ab and FH Ab, depending on the blocking reagents used in the tests. The authors should mention what measures were taken to exclude false positivity in their patients.

Minor revisions

Author Response

We thank the reviewer for his valuable suggestions and comments, which we are convinced will contribute to a significant improvement in the quality of our manuscript. Our point-by-point answers are presented below.

Reviewer 2 Report

This study retrospectively analyzed 7 patients with antibody-mediated thrombotic microangiopathies (TMAs), including 5 patients with atypical hemolytic uremic syndrome (aHUS) due to antibodies against complement factor H (CFH) and 2 patients with thrombotic thrombocytopenic purpura (TTP) due to antibodies against ADAMTS13. The patients were treated at a single center with a protocol consisting of plasma exchange, steroids, cyclophosphamide, and rituximab. Most patients achieved sustained remission without maintenance therapy after a median follow-up period of 11.7 years.

In the 5 aHUS patients, plasma exchange was used to achieve initial remission. 4 of these patients also received early immunosuppressive therapy with steroids, cyclophosphamide, and rituximab. These 4 patients experienced sustained remission with preserved kidney function. The patient who did not receive early immunosuppression progressed to end-stage renal disease. The 2 TTP patients were treated successfully with the same immunosuppressive protocol and remain in remission. This study provides evidence that plasma exchange along with early immunosuppression can achieve durable remissions in antibody-mediated TMAs. However, the small sample size limits definitive conclusions and further research is needed to confirm the optimal treatment approach.

Constructive points on how to improve this study:

  1. The sample size is very small (n=7), making it difficult to draw definitive conclusions. A larger sample would allow for more robust statistical analysis.
  2. There is no control group for comparison to the treatment group. Including a control or comparison group could strengthen the evidence for the efficacy of the treatment protocol.
  3. The study design is retrospective and observational. A prospective, randomized controlled trial would provide higher quality evidence.
  4. Details of the treatment protocol and dosing are sparse. Providing more details would improve reproducibility.
  5. Longer follow-up of patients would help better determine long-term outcomes like kidney function.
  6. The role and necessity of each component of the immunosuppressive regimen is unclear. Further study on optimal treatment combinations could help refine the protocol.
  7. Objective criteria for remission should be defined, rather than stating "clinical remission."
  8. Statistical analysis is limited. More rigorous statistical approaches could identify significant relationships in the data.
  9. The study population is quite heterogeneous, including aHUS and TTP patients. Focusing on one disease may allow stronger conclusions; or may provide subgroup analysis
  10. Mechanistic studies on how the treatment works would be informative. Assessing antibody levels over time may provide insights.
  11. constructive points on how to improve this study:
  12. The sample size is very small (n=7), making it difficult to draw definitive conclusions. A larger sample would allow for more robust statistical analysis.
  13. There is no control group for comparison to the treatment group. Including a control or comparison group could strengthen the evidence for the efficacy of the treatment protocol.
  14. The study design is retrospective and observational. A prospective, randomized controlled trial would provide higher quality evidence.
  15. Details of the treatment protocol and dosing are sparse. Providing more details would improve reproducibility.
  16. Longer follow-up of patients would help better determine long-term outcomes like kidney function.
  17. The role and necessity of each component of the immunosuppressive regimen is unclear. Further study on optimal treatment combinations could help refine the protocol.
  18. Objective criteria for remission should be defined, rather than stating "clinical remission."
  19. Statistical analysis is limited. More rigorous statistical approaches could identify significant relationships in the data.
  20. The study population is quite heterogeneous, including aHUS and TTP patients. Focusing on one disease may allow stronger conclusions.
  21. Mechanistic studies on how the treatment works would be informative. Assessing antibody levels over time may provide insights.

Author Response

Thank you very much for acknowledging of our research data. In accordance with your suggestions, we have revised the paper and supplemented the limitations section wich we are convinced contributed to the improvement of the quality of our manuscript. Since the comments and suggestions for improvement largely overlap, we will answer them together. Please find our point to point responses below.

Round 2

Reviewer 2 Report

all of comments have been addressed